# Development of the Concept of Space up to Newton

Danilo Capecchi 

Structural and Geotechnical Engineering, University "La Sapienza", I-00184 Rome, Italy;
danilo.capecchi@uniroma1.it; Tel.: +39-(06)-45448007

**Definition:** The concept of space, ubiquitous among all humans from birth, has changed profoundly in the course of the history of Western civilization, the only one to be considered here. An important contribution to this change was the theoretical elaborations of the philosophers of nature and mathematicians, started in Ancient Greece. Here, the process is considered up to Newton, when the concept of space for physicists, who then replaced the traditional philosophers of nature, took on a connotation that remained substantially undisputed for two centuries—that of absolute space.

**Keywords:** space; mathematics; natural philosophy; void; atomists; Patrizi; Gassendi

## 1. Introduction

There are countless studies on the history of the concept of space (see, for instance, the following monographs on the general characteristics [1–9]). This notwithstanding, here, I attempt to offer a new contribution. The present work is not, strictly speaking, historical research, in the sense that there is no attempt to search for documents not yet known among modern historians of science. The aim is rather to present a new point of view on the meaning of already published documents and the role that they have played in the ideas of modern physicists and mathematicians, especially regarding the study of motion. This paper is a substantial historical survey of prominent views, although primary sources are generally used.

Space is a term that today has many meanings [10] (pp. 1–3), and it can be considered from different perspectives [11]. In this entry, the points of view of natural philosophy and mathematics are mainly considered; abstraction is made, as far as possible, from epistemological, anthropological, psychological/sociological or theological aspects, always present in elaborations of ancient times.

The first conceptions of space were intuitive ones that even a modern child could describe: an environment outside of ours, populated by gods, ghosts, and material objects, and one has vague notions of their amplitude, depth, distance, and direction. The first mathematicians, particularly the first Greek mathematicians, had only an intuitive idea of space, as far as is known from existing written records, most of which have not reached us. However, they understood the operations of space by means of solid geometry. In fact, even architects in designing their buildings, and soldiers with their weapons, had an intuitive idea of ambient space that was sufficient for their purposes, and astronomers operated in space; even if they did not pose, perhaps, ontological and epistemological questions about space, they raised at least the question of the finitude of the universe and the substance of which it is constituted: vacuum or plenum.

Natural philosophers and mathematicians—both considered in a broad sense, to include most people that today would be classified as scientists—saw space in different ways. The natural philosopher had to answer general questions concerning the existence of material and immaterial beings, creation, and the presence of God; the mathematician, at least until the 18th century, was essentially interested in space as an environment in which to build geometric figures and study the motion of bodies.

A precise notion of space was not a pressing need for the mathematicians before Newton. All considered space was not of interest in itself; the role that it played in the motion of bodies was relevant. The pioneers of early modern science, such as Galileo, for example, did not need to go beyond the intuitive idea of space to study the motion of bodies in a terrestrial environment. It was sufficient to possess a metric and a reference frame, and the Earth was enough for this, even if it was known that it moved within the 'space'. This notwithstanding, the study of the analyses that the philosophers of nature carried forward through two millennia of history is very important for the understanding of how mathematicians have utilized this concept since the early modern era.

The purpose of this entry is not only to show how the concept of space developed in history in the reflection of natural philosophers, which is certainly interesting in itself, but mainly to show that it was fundamental in the development of classical mechanics and the entirety of modern physics.

1.  First, the various conceptions of space in Greek antiquity are presented, comparing them with the intuitive idea of space;
2.  Then, the harmonization during the Renaissance of the notions of scholasticism and the rediscovered Ancient Greek sources are considered;
3.  Finally, the vision of absolute space in the modern era is presented, successfully overcoming the intuitive idea of space, made necessary for the development of a science of motion with cosmic significance, with Newton and Gassendi.

## 2. Relationalist and Substantivalist Approaches to Space

In traditional natural philosophy, as well as in modern physics, we generally assume a realistic point of view—that is, our ideas of space faithfully reproduce entities that in fact exist outside of us. In such a case, there are two possible means of looking at space that move beyond immediate perception. On the one hand, there is (a) what is now called *relationalism*, which denies the existence of space as an object in its own right, seeing it rather as relations between material objects; this leads to what, in the following, will be named *internal space*. On the other hand, there is (b) what is now called *substantivalism*, which sees space as an existing being, over and above the other material objects and processes of the world. By canceling a body, the space that it occupies becomes void, similarly to the notion of a void as the space where there are no bodies; this leads to what, in the following, will be named *external space*.

In Albert Einstein's words,

These two concepts of space may be contrasted as follows: (a) space as positional quality of the world of material objects; (b) space as container of all material objects. In case (a), space without a material object is inconceivable. In case (b), a material object can only be conceived as existing in space; space then appears as a reality which, in a certain sense, is superior to the material world. Both space concepts are free creations of the human imagination, means devised for easier comprehension of our sense experience [1] (Foreword by Albert Einstein).

A particular view of substantialism gives rise to the concept of *absolute space*, a term introduced by Newton to indicate an external space, infinite, isotropic, uniform, perfectly penetrable, and immovable, where he located motion and studied its variation. The fruitfulness of Newton's absolute space silenced, for several centuries, the objections raised by the ontology of an external, pre-existing entity and space of type (b) that was generally accepted by scientists. Newton's decision was, in the contemporary state of science, the only fruitful one. However, the subsequent development of physics has shown that the resistance of Newton's 'colleagues', Leibniz and Huygens, supporters of a relationalist view (well documented in [12,13]), intuitively founded but supported by arguments considered as inadequate at the time, was actually justified; see, for instance, [5,14].

A victory over the concept of absolute space could today become possible only because the concept of the material objects has been gradually replaced by that of the field—under the influence of the ideas of Faraday and Maxwell. If the laws of this field are, in general,

not dependent on a particular choice of coordinate system, then the introduction of an independent (absolute) space is no longer necessary [1] (Foreword by Albert Einstein).

It is generally believed that the study of the nature of space was mainly carried out by philosophers rather than mathematicians. This is true, albeit partially only for the modern era, until the 18th century. In ancient times the difference between mathematicians and philosophers was more nuanced, and mathematicians played a role at least as important as philosophers, also because that of space is from many points of view a concept that cannot be separated from mathematics. Moreover, since the 19th century, mathematicians have been deeply involved in the development of the concept of space.

Unfortunately, there are no studies on the role played by ancient mathematicians on the influence their geometry has had on the concept of space. Generally it is sustained, see for instance [15,16] (Introduction), that at the beginning mathematicians focused their attention on figures, rather than to the ambient space; thus, theirs was a geometry of figures; only later was space considered explicitly as an independent concept.

Alexandre Koyré in his famous text *From the closed world to the infinite universe* of 1957 [7] attributes an important role to Euclidean geometry in forming the idea of infinite space, and he seems quite convincing. Edward Grant, on the other hand, argues that in no case did the discussions on the nature of space and its finiteness refer to Euclidean geometry, from Archytas of Tarentum to Pierre Gassendi [2] (p. 108).

The point of view of Francis M. Cornford (1874–1943), a British professor of ancient philosophy, suggests a thesis close to that of Koyré but more radical, which although not supported by historical evidence, is very fascinating [4]. Cornford argues that the ancient conceptions of space, even the Greek ones at least up to the 5th century BCE, were related to a cosmos of finite size and spherical shape. Greek geometry, which had been taking shape with Thales and Pythagoras starting from the 6th century BCE, was not based on the circle but rather on the straight line. As mathematicians formalized their geometry and defined their concepts, especially that of straight lines, and discussed their properties, they naturally came to propose a mathematical space, independent of the nature of real world, which was naturally limitless. In fact, what sense had the postulate of parallels if one did not conceive of an unlimited space?

At the moment it was essentially a purely geometric space, defined by an axiomatic very distant from common sense. However, mathematicians could not fail to see that their ideal entities had counterparts in the material world. Moreover, mathematicians were generally aware of the discussions on the concept of space brought before their contemporary philosophers or those who had preceded them. One of the first mathematicians to draw the conclusions from a geometry not yet fully formalized to the physical space was Archytas of Tarentum (b. 430 BCE). He is famous for an argument, at least as handed down by Eudemus of Rhodes (b. 350 BCE), in which he spoke of an infinite void space conceived as a container of matter.

> If I am at the extremity of the heaven of the fixed stars, can I stretch outward my hand or staff? It is absurd to suppose that I could not; and if I can, what is outside must be either body or space. We may then in the same way get to the outside of that again, and so on; and if there is always a new place to which the staff may be held out, this clearly involves extension without limit. Now if it is a body, the assumption is proved; if it is space, *given that space is what a body is or can be in* [emphasis added] and that when we speak of eternal substances we must certainly say that it is what it is in potentiality, even so it will be demonstrated that infinite are body and space [17] (p. 267. Simplicius, Physica 467, 26. Translation into English in [4]).

However, according to Cornford were the atomists, with the 'mathematician' Democritus, who projected the concepts of mathematics into natural philosophy, conceiving physical space as a pure void and infinite container populated by atoms. Basically, Cornford argues that the idea of space that we call Euclidean was the result of a cultural revolution carried out by Greek mathematicians, a revolution against traditional conceptions. The physical

ideas of the atomists found a strenuous resistance from contemporaries who opposed a strongly rooted conception of cosmos. Unfortunately one of the opponents was Aristotle—a staunch defender of the traditional conception of the finite spherical cosmos—whose philosophy will be dominant from the 12th to the 16th centuries. Only the rediscovery of Greek texts, the development of Neoplatonism, the discovery of Lucretius (15th century), who gave a complete account of Epicurus atomism, and of Diogenes Laertius (16th century), who carried forward the conceptions of the most ancient atomists, allowed the recovery of the atomistic conception of space in the early modern era.

According to Cornford the revolution started by the Greek mathematicians, after the introduction of non-Euclidean geometry in mathematics and the theory of relativity in physics, developed at the turn of the 20th century, is now opposed by another revolution, or perhaps a counter-revolution, which leads again to the idea of a finite world in which the 'circle' and not the straight line becomes fundamental again.

## 3. Concepts of Space in Ancient Greece

The reflections on space of the Greek philosophers and mathematicians were fundamental for Western civilization, the only form of civilization considered here. These dated back to ancient times, and has antecedents at least in the middle east: Egypt, Mesopotamia for instance; some hints of this influence can be found in [18]. For our purposes, however, it has been considered sufficient to start from the 5th century BCE and bring back the considerations that were then elaborated in the modern era. Here, one finds the positions of the atomists who conceived of infinite space as a void populated by full, very small, rigid atoms, then there are the conceptions of Plato and Aristotle who thought the universe as a plenum of finite dimension and, finally, those of the Stoics who postulated a finite cosmos immersed into an infinite void space. The ideas of the above philosophical schools, though very different from each other, were part of a shared knowledge, in the sense that the later philosophers knew, at least partially, the ideas of the predecessors.

### 3.1. Substantivalist Views of Space

Atomists and Stoics, presented below, both conceived space as a receiver. However, the atomists only paid a great deal of attention to motion paying also attention to kinematics aspects. Space beside being what contains atoms (matter) was also the place where their rectilinear and uniform motion occurred.

### 3.1.1. The Atomists

As for the atomists a brief reference to only Democritus and Epicurus is made, meaning by these names not so much historical characters but rather labels, of ancient atomism in the first case and in the second the synthesis of the new atomism mainly for how it is reported in Lucretius's *De rerum natura*.

Democritus considered space as external unlimited, empty, populated by invisible hard atoms that are infinite both in number and in form. It should be noticed that according to some interpretations Democritus probably could not conceive atoms immersed in an empty space as a container but rather he thought about an alternation of void and atoms; where the void had some form of reality for Democritus, and both atoms and void were in a place [1,19] (p. 11; pp. 179–182). Johannes Stobaeus (after 5th century CE) stated moreover: "And he [Democritus] said that it was possible for an atom to be as big as a world" [20] (p. 88). Note that Democritus did not use the very word atoms, but rather solids, beings, substances; see [21] (p. 7).

> Democritus thought the nature of the eternal things consisted of small substances infinite in number; these he placed in space, *separate from them and infinite in extent* [emphasis added]. He called space by the following names: *the void*, *nothing*, and *the infinite*, and each of the substances he calls *thing*, *the solid*, and *what is*. He thought that the substances were so small as to escape our senses. They have all kinds of forms and all kinds of shapes and differences of size. Now

from these as elements the visible and perceptible bodies are genera. He says that they conflict with one another and travel about in the void because of their unlikeness [20] (pp. 70–71. Simplicius, commentary on *De caelo*).

In the above quotation Simplicius assumed that space, void, and nothing are synonymous. Actually, things are much more complex, or, alternatively, not very clear. Void (or nothing) had some degree of reality for Democritus; 'nothing' is not really nothing but something of real, something that can possibly occupy space, and the debate on this subtle concept—and consequently on that of space—has been in progress since the time the atomist assumed the existence of void. In any case, even if there was a difference between space and void the two concepts inevitably flow together, and Simplicius's flaw is not simply a misunderstanding.

Atoms move in every direction, courting each other until they collide. There is no explicit reflection on the nature of space in the above quotation; the fundamental concept is in fact that of motion, considered real (absolute) without posing any problem about its origin. However, even if not specified, it seems reasonable that Democritus thought of an infinite space—because it has to contain infinite atoms— as isotropic, uniform like Newton's absolute space. However, even if this were not his thoughts, the reading of his writings by early modern philosophers and mathematicians gave the impression of an absolute motion in an absolute space.

Space, atoms, and motion of Epicurus are different. Here, the space is explicitly said to be the container of all atoms. However, it is not isotropic like Democritus', but  polarized. There is a prevailing direction that defines an up and a down. Atoms are brought down by their 'weight', move in a straight line all with the same speed [22]. They would continuously move downward, as happens to drops of water in the rain, if a spontaneous deviation did not intervene, known as *clinamen* a Latin word Lucretius used [23] (Book 2, 290–293). The clinamen leads atoms to collide with each other, determining the possibility that they aggregate to form bodies: "The atoms, as their own weight bears them down in straight lines through the void, at a absolutely indeterminate times, in indeterminate places, they decline from their course just a little, so to speak that trend has changed" [23] (Book 2, 217–220. My translation).

3.1.2. The Stoics

The conception of space of Stoics is important not only in itself, but also, and mainly, because it was known in the Middle Ages and in the Renaissance, much more than that of the atomists and it was the occasion for the philosophers of the time to discuss the idea of a void and infinite space. Much of the cosmology of the Stoics followed the most widespread ideas in antiquity also shared by Aristotle: the cosmos is a finite and stable structure, a plenum filled with *pneuma* and elements, such as earth, water, air, and fire, surrounded by an infinite void; the heavens are homocentric spherical revolving bodies made of a subtle substance (primeval fire?); the earth is spherical, and immobile at the center of the space. Cosmos is in itself necessarily limited, whereas what is outside it is a void space that extends without limit in every direction. Of this space, the part that is occupied by body is called a place, while that which is not occupied is simply void.

Stoics justified the existence of void both with arguments from physics (a) and from logic (b).

(a)     Every body is necessarily present in something; but the thing that a body is present in, given that it is incorporeal and, as such, without physical contact, must be distinct from what occupies and fills it; we therefore speak of such a state of subsistence— namely, a capacity to receive body and be occupied as void [24] (p. 22).

Cosmic order is not eternal but must eventually perish. Its destruction comes in a tremendous conflagration, called the *ekpyrosis*, in which everything is changed into fire. Then, after a period during which nothing but fire exists, the world order will again come into a new existence. The future cosmos will be identical to this one in every respect; there will be another Plato and another Socrates; Socrates will marry

another Xanthippe, etc. The history of the cosmos will proceed in eternal cycles of destruction and restoration [25] (p. 185). Though it was not strictly necessary the expansion of the cosmos called for a void space outside it, it may suggest it.

(b) The logical justification for the existence the void follows one of the definition derived from Aristotle, for which void is that "in which the presence of body though not actual is possible" [26] (I.9 279a). According to Aristotle logic a proposition is possible only if it becomes actual at some time. According to the Stoics this condition is instead not necessary; a proposition is possible if nothing external prevents it from being true. Consequently, the condition beyond the periphery of the cosmos satisfies the definition of void, even if body never comes to occupy it [25] (p. 106).

The infinity of void also had physical (a) and logical (b) justification.

(a) It is absolutely necessary that this void extend without limit in every direction from the cosmos, as we may learn from the following (principle): everything that is limited has its limit in something different in kind from it (This is the same argumentation of the atomist-Epicurean tradition, see [23] 1.958). To take an obvious example: in the whole cosmos air, because it is limited, ceases (to be air) at two bodies different in kind, aether and water. Similarly, the aether ceases at both the air and the void, the water at both the earth and the air, and the earth at the water. It is, then, necessary that if the void enclosing the cosmos be limited rather than unlimited, it ceases to be a void and is something different in kind. However, it is no different in kind from the void, and there is no point at which the void ceases that can be conceived of. Therefore, the void is unlimited [24] (pp. 29–30).

(b) According to Chrysippus, just as non-being has no limit, there is no limit to non-being, of which non-being is the kind of thing that void is. In its own subsistence, it is then infinite [27] (p. 426).

### 3.2. Relationalist Approaches to Space

Strictly speaking, between Aristotle and Plato considered below, only the former had a relationalist view of space, which reflected his conception of geometrical object as simple abstractions. Plato gave geometry the pre-eminence over matter. His was a geometrical space; however having a local character, that is being scarcely interested to conceive space as the container of the whole world, Plato's concept does not differ much from Aristotle's. Both philosophers gave a fundamental role to local (spatial) motion to explain changes in the worlds, but when motion was described kinematically they viewed space in an intuitive way, like the common men.

### 3.2.1. Plato and the Timaeus

Plato in the *Timaeus* conceived the sensible world, the cosmos, as a plenum, spherical, and finite. There is some uncertainty about where Plato placed his finite universe. According to Pierre Duhem, it is surrounded by an infinite space, which he identifies with the void [6] (vol. 1, pp. 37–38).

Plato considered that there is one kind of being which is always the same, uncreated and indestructible, never receiving anything into itself from without, nor itself going out to any other, but invisible and imperceptible by any sense, and of which the contemplation is granted to intelligence only (what we know as ideas). Additionally, there is another nature of the same name with it, and like to it, perceived by sense, created, always in motion, becoming in place and again vanishing out of place, which is apprehended by opinion and sense. Additionally, there is a third nature, which is space (χώρα), and it is eternal and admits not of destruction [28] (52a).

> In the same way that which is to receive perpetually and through its whole extent the resemblances of all eternal beings [space] ought to be devoid of any particular form. Wherefore, the *mother and receptacle* [emphasis added] of all created and visible and in any way sensible things, is not to be termed earth, or air, or fire, or water, or any of their compounds or any of the elements from

which these are derived, but is an invisible and formless being which receives all things and in some mysterious way partakes of the intelligible, and is most incomprehensible [28] (49a).

There is in the above quotation a specification of space: it is mother and receptacle, or in other word matter and space. There has been a large debate among commentators about the meaning of space (χώρα). For a list of references see [29] (pp. 29–30); in any case space is not a nothing, it has properties, and some form of activity like a living aether.

Space is the container of perishing beings formed by very small atoms, as for Democritus, with a very simple geometric structure however, a shape for each of the four elements postulated by Empedocles: earth, water, air, and fire, instead than an awful variety of shape and size. The earth is assigned atoms of a given cubic shape, which have a greater difficulty in flowing among them. The other elements are assigned solids with equilateral triangular faces equal to each other; the polyhedron which is the most mobile or the tetrahedron (4 equilateral triangular faces) is attributed to fire, the icosahedron (a solid with 20 faces) is attributed to the water and air which is of increasing intermediate mobility between water and fire is attributed the octahedron (two superimposed tetrahedra, 8 equilateral triangular faces).

The space of Plato resembles that of Descartes; he (Plato), although the exposition of the *Timaeus* is cryptic, did not introduce an explicit concept of matter; his atoms are only boundaries of space and in a certain sense space coincides with body. Compared to Descartes, Plato's structure is simpler; instead of corpuscles of any shape there are atoms of only four types, furthermore space has an existence by itself, space is matter while in Descartes matter is space. In Plato, as in Descartes, motions imply whirlpools.

### 3.2.2. Aristotle and the Place

Aristotle's universe, which he called *Heaven*, is finite and spherical in shape and is a plenum in which void has no place. More precisely, it is defined by three concentric spheres, the first outermost sphere is the sphere of the stars or *first mobile*, the second sphere at the height of the moon delimits the sublunar world, the third sphere encloses the earth. The earth is at rest in the center of this universe, while the first mobile rotates with diurnal motion. There is nothing beyond the first sphere. By nothing it is meant that it makes no sense to ask what is there, in particular there is no empty space.

Notwithstanding the relationalist view of Aristotle who denied any existence of an external space independent of bodies, for a modern reader his universe exhibits some characteristic of absolute (for instance immovability), albeit finite, space. By making a base on earth, any motion could be monitored and qualified as absolute, because the earth is at rest. Moreover, it is possible to speak of up and down and give justification of the concept of natural places. Even the Greek astronomers, who carried out measurements of angles, could be of this opinion, although there are no critical discussions on the matter.

However, Aristotle did not think so. He gave a vague definition of (three-dimensional?) space only in the *Categoriae*, when he talked about quantity.

Space, likewise, is a continuous quantity; for the parts of a solid occupy a certain space, and these have a common boundary; it follows that the parts of space also, which are occupied by the parts of the solid, have the same common boundary as the parts of the solid. Thus, not only time, but space also, is a continuous quantity, for its parts have a common boundary [30] (6.5a).

However, in his technical writings on natural philosophy, in particular in the *Physica*, he gave a local definition only, using the notion and the word of *place*. To understand what place is according to Aristotle, one must think that he conceived of a full universe without a vacuum, therefore every body is always contained in another body. The place is the "boundary of the containing body at which it is in contact with the contained body (By the contained body it is meant what can be moved by way of locomotion)" [31] (4.1.212a) and Aristotle explicitly excluded that the place is the spatial extension (the volume) of the contained body, otherwise it would be a body itself.

Place, however, is something ontologically different from the material boundary of the contained body:

> The existence of place is held to be obvious from the fact of mutual replacement. Where water now is, there in turn, when the water has gone out as from a vessel, air is present. When therefore another body occupies this same place, the place is thought to be different from all the bodies which come to be in it and replace one another. What now contains air formerly contained water, so that clearly the *place or space* [emphasis added] into which and out of which they passed was something different from both [31] (4.1.208b).

This passage provides an idea of the Aristotelian conceptions of space; it is other than body, but it has no reality of its own. Places take precedence of all other things. For that without which nothing else can exist, while it can exist without the others, must needs be first; for place does not pass out of existence when the things in it are annihilated [31] (4.1.209a).

Motion is simply a change of place; it is thus a relative motion. However, it is also absolute, as it can always referred to the earth which is immovable in its natural place and thus, as already noticed, offers a natural reference for an absolute motion.

Aristotle's local definition of space, however, comes in partial contradiction with the idea of natural places that plays an important role in his physics. Inside the sublunary sphere, there are substances that are born and die, and, as for Plato, they are formed of the four elements: earth, water, air, and fire; beyond the sublunar sphere there are bodies formed by the fifth element, the fifth essence or aether, which is incorruptible and eternal. Each of these elements has its own natural place toward which it tends in the sense that if an element is outside its natural place it is in an unstable equilibrium situation and tends to move with local motion to reach its natural place. Thus, a non-uniformity is introduced into the collection of places; they are not all the same and those above are different from those below.

Though Aristotle conceived his universe as a plenum and denied the existence of void, he gave a definition of it, which influenced the discussions on space in the Middle Ages; below one of the most influential.

> The void is thought to be place with nothing in it. The reason for this is that people take what exists to be body, and hold that while every body is in place, void is place in which there is no body, so that where there is no body, there must be void [31] (4.7.213b).

That is Aristotle had in medieval terms formulated the accepted *quid nominis* or definition of vacuum [2] (p. 10).

## 4. Concepts of Space in Middle Ages and Renaissance

In the Middle Ages, especially starting from the 13th century, the conception of space was that of Aristotle; the Stoic and mainly the atomistic ideas, although not completely unknown, were little explored by most of the philosophers who read together with Aristotle his ancient, Latin and Greek commentators.

The role played by mathematicians in the development of the concept of space in this period is quite clear: they gave no fundamental contribution. First, there was no longer a mathematical school as important as that of ancient Greece; second, the interests of mathematicians (including in this category not a few scholars today considered as philosophers) were of a more technical nature since the frame of reference of mathematics and geometry, in particular, was substantially well defined. The traditional philosophers, for a long time associated with scholasticism, had to do with Aristotelian philosophy of nature and were harnessed by its conceptions of space and void. They developed acute and even very interesting arguments, from a philosophical point of view, but as long as they became entangled in the canvas in Aristotelianism, at least until the 16th century, they did not develop conceptions of space that could be used by mathematicians to mastering the problems of motion previously studied in natural philosophy. Only with Huygens, Newton,

and Leibniz, essentially the three greatest mathematicians at the turn of the 18th century, did the concept of space cease to be the occasion for partially inconclusive discussions for the development of modern mechanics.

Medieval philosophers were aware of the criticalness in Aristotelian conceptions of space, connected to the definition of place. This definition led to a number of contradictions. One of the most important of them concerned the justification of the rotation of the first mobile: what is the place of this sphere if there is nothing outside it? What is the point of talking about rotation if there is no place to refer to it?

There were two approaches to overcome this impasse, one cosmological and the other ontological. The cosmological approach was possibly simpler to follow; one could postulate, for example, another more external sphere, this time fixed, the empyrean outside of which there is God. However, this led to a closed circle: what was the place of this external sphere? As an alternative, one could admit the existence of an infinite being (space). An important contribution in this direction was given by the heliocentric theory of Nicolaus Copernicus reported in *De revolutionibus orbium coelestium*, published in 1543 [32]. According to Copernicus, the 'first mobile' was, in effect, immobile, that is, the sphere of the stars was fixed as the boundary of a finite cosmos, which had the sun at its center. Although Copernicus remained anchored to the hypothesis of a 'material' sphere that keeps the stars fixed, the road toward the conception of an infinite universe begins to make its way, supported by new astronomical observations which suggested that it was difficult for the stars to be all equidistant from the center of the universe [33].

The ontological approach was more difficult to pursue but more fruitful. Before and also during the Renaissance most philosophers lended toward an internal view of space, in the footprints of the Aristotelian tradition which denied void, also because little was known about the writings of predecessors. In particular, the writings of Ioannes Philoponus (5th–6th century CE) were not known, at least in Latin, who was probably the only one to lend toward an external view of space (see [2] (p. 19)). The internal ontological view of space started to be questioned at the dawn of Renaissance as soon as the infinity of space was admitted, even though at the beginning it was only a possibility. A turning point was made possible only thanks to the new interest in the works of the Stoics and atomists of ancient Greece. The atomistic ideas, although known through ancient commentators, found their full expression in Lucretius *De rerum natura*, published in 1473 after being rediscovered by the Italian humanist Gian Francesco Poggio Bracciolini (1380–1459).

To signal the interesting exception to the internal view of space in the Middle Ages represented by the Jewish philosopher Hasday Crescas (1340–1410) and his *Or adonai* (The light of the Lord) a historical and critical investigation of the main problems of Aristotle's *Physica* and *De caelo*, leading to an external view of space. His work is interesting not only in itself, as it represents an alternative to the dominant view of the Middle Ages, but mainly because it was read in the Renaissance; it influenced for instance Giovanni Pico della Mirandola (1463–1494), who named him explicitly, and Giordano Bruno. The last did not name Crescas, but his argumentation against Aristotle positions on void, place, the existence of infinite worlds, etc., are the same as Crescas' [34] (p. 25). It is difficult to evaluate the real influence of the Jewish philosopher on his readers. Independently of his position on specific points, his critical attitude toward Aristotle was for sure an important step in the history of Western philosophy. Crescas criticized Aristotle's idea of place, by assuming for it the volume occupied by a body instead of its exterior boundary.

> The truth of matter, as it seems, is that the true place of a thing is the interval between the limits of that which surrounds [...]. *That it should be so it can be also shown from the consideration that place must be equal to the whole of its occupant as well as to [the sum] of its parts* [emphasis added] [34] (pp. 195–199).

Notice the request of additivity (a logical need?) for place in the last part of the quotation. That is that the place of the union of two places equals their summation. Place as a surface cannot, in general, satisfy this requirement, place as a volume can.

Crescas' ideas show a striking resemblance to those of Philoponus, but differently from him he believed in an infinite void space, a cosmology very akin to the Stoics, even though his space allowed for the possibility of other worlds beyond ours "thus every thing said in negation of the possibility of many words is vanity and a striving against wind" [34] (p. 217). Below his argumentation in favor of the infinitude of space, very similar to that of atomists and Stoics:

> Since the void has been shown to be a magnitude, it has thus been shown that an incorporeal magnitude exists. However, this incorporeal magnitude outside the world cannot have a limit, for if it had a limit it would have to terminate either at a body or at another void. That it should terminate at a body, however, is impossible. It must therefore terminate at another void, and so it will go on to infinity. It has thus been shown that on their own premises an infinite incorporeal magnitude must exist [34] (p. 189).

*Francesco Patrizi and Giordano Bruno*

With Renaissance many philosophers thought about the real infinity of space. Among them the neoplatonist philosopher Fancesco Patrizi (1529–1597), a proponent of the stoic conception of the universe, who suggested a clear and new ontology of space.

The role attributed to Patrizi in spreading the concept of (absolute) space is controversial. According to some historians supporting continuity in the history of science—such as Edward Grant and Pierre Duhem, for instance—Patrizi's role was fundamental, much more than that of Giordano Bruno, for instance. According to this view, Patrizi based his analysis in continuity with medieval philosophy, either Platonic, Stoic, or Aristotelian. The supporter of the discontinuity thesis—such as Alexandre Koyré for instance—saw the affirmation of a new concept of space in the rediscovery of Greek authors, the atomists in particular, and saw the role played by Bruno more interesting than that of Patrizi. In any case the role played by Patrizi is relevant and worth of an accurate study, at least for priority in time.

Patrizi spoke in depth about his ideas on space in the first three books of the *Pancosmia—De spatio physico, De spatio mathematico, De physici ac mathematici spacii, affection nibus*—a section of his opus magnus, *Nova de universis philosophia*, published twice in 1591 and in 1593 [35]. The first book of the *Pancosmia* introduces space since the beginning. It is what God created first, before all the other things: "Now what did the Supreme Maker create before all other things apart from Himself? [...] However, this is Space itself. For all things, whether corporeal or incorporeal, if they are not somewhere, are nowhere; and if they are nowhere they do not even exist" [35] (f. 61r, translation into English in [36]). Space is an indivisible whole, because Divinity is as a whole indivisible and it exists in space [35] (f. 61v).

Here, it is clearly stated that space was created by God (out of nothing and in some instant of time?). According to [37] (Chapter 1). on the other hand, more than a creation it would have been an emanation, following the Neoplatonic philosophy. Additionally, it would have been an emanation for what concerns light, heat, and fluid (*lumen, calore, fluore*) also [35] (f. 83v). Beings that gradually (in time), while remaining incorporeal, come to existence and approach corporeality.

Ascertained the existence and priority of space, Patrizi wondered about its ontological nature; in this he remained in some way tied to the Aristotelian categories, if only to deny them:

> First space is a three dimensional being. It is something incorporeal having all the three dimensions of a body, though it is not a body. [...]. Space therefore is substantial extension (extensio hypostatica), subsisting per se, inhering in nothing else. It is not quantity. And, if it be a quantity, it is not that of the categories, but prior to it, and its source and origin. Nor can it be called an accident, for it is not the attribute of any substance. [...]. For all these reasons, therefore, it is very clear that Space is above all a substance, but not the "substance" of the

category. [. . . ] What then is it, a body or an incorporeal substance? Neither, but a mean between the two [. . . ] Therefore it is an incorporeal body and a corporeal non-body [35] (f. 65r-v. Translation into English [36]).

The introduction of a corporeal and incorporeal being could appear as inconsistent to a modern reader, but in Patrizi's time it was a concept well founded in the Neoplatonic circles, and thus from certain point the use of this wording should not to be considered as a play of words.

To these ontological considerations, Patrizi added others that we could qualify as historical. That space exists and is something is certified by fact that it has generated terms like: dimension, distance, interval, space, place [35] (f. 61v).

A further characterization of space is obtained by considering the presence in it of bodies; of them Patrizi claimed that no one can doubt they are in place (locus) since they are surrounded by space on all sides [35] (f. 61r). He used the term *locus*, but criticized the meaning that Aristotle gave to it, who was contradictory because after asserting its three-dimensional character, then he reduced it to bi-dimensionality [35] (f. 62r).

After a series of arguments on the concept of place and its three-dimensional nature, Patrizi concluded: "Hence locus, not being a body, will of necessity be a space provided with three dimensions length, breadth, and depth-with which it receives into itself and holds the length, breadth, and depth of the enclosed body" [35] (f. 62v). To complete the characterization of space the problem of its relationship of space with the vacuum is examined. Patrizi's vacuum was not the absolute emptiness but simply the absence of corporeal substances. The vacuum was in fact a plenum, filled with light (then heat and fluid).

According to Patrizi the void can exist as a huge space only outside the cosmos, Inside the cosmos it can exist only in the form of small spaces and indeed it necessarily exists inside bodies as a micro space [35] (f. 63v).

Moreover, it is a plenum to the senses and in popular parlance, but according to reason, as I have shown, it is a vacuum, and its being a plenum does not pertain to its essence, but is rather accidental to it. For plentitude, of course, comes to it from the bodies, which are different from it in nature in such a way that the spaces which we spoke of before as essential attributes of bodies, are for the most part accidental to them, their own essence consisting in resistance [35] (f. 64v. Translation into English in [36]).

Patrizi argued on the existence of an *extra-modan* void with logical, ontological and historical considerations, recalling the various conceptions of the philosophers of the past, in particular of the Stoics. Having proved the existence of an extra-worldly he also proved that it is infinite with arguments both logical and theological [35] (f. 83v).

However, does the space occupied by bodies have the same nature as empty space? Or has the space which holds the cosmos, which is its locus, the same nature of the outer space? Patrizi's answer is positive [35] (f. 65v).

Basically, for Patrizi, space is entirely unmoved and unmovable, infinite, and uniform. However, the ontological uniformity is broken by its stoic cosmology. For Patrizi space has a center that coincides with the center of the cosmos: "the centre of the space therefore is at the midpoint of the infinite universum space" [35] (f. 64v). The uniformity of space is also broken by the attempt to conciliate his cosmology with Aristotle's. Indeed at the end of the book devoted to physical space, Patrizi suggested the possibility that not all the part of space of the finite cosmos are equivalent.

This difference, arising as it does from located bodies, is accidental to them, unless it should be proved that those parts of Space were so arranged from the beginning that the one holding the earth is incapable of holding the air, and the one holding the water is unable to hold the heaven, air, or earth, and that each part received bodies peculiar to itself [35] (f. 65v. Translation into English in [36]).

Considering space endowed with a center not necessarily implies breaking off uniformity. Indeed once the existence of a single world is given for granted, it is quite natural to imagine the surrounding space as an infinite sphere. It could be simply an analogy. The problem of the acceptance of the qualitative difference of the space occupied by different sort of bodies (earth, water, fire) is more difficult to justify, even though this is considered only a possibility.

Once defined the structure of space, Patrizi went on to present how its structure can be understood; the way is provided by mathematics, which becomes the fundamental discipline of natural philosophy:

> Carrying the same reasoning further, it is clear to anyone undertaking the study of natural things that the science of Space [geometry] must be acquired and taught before either natural science or that treating of the actions and passions of men. For the latter come after the activities of nature, and these in turn come after Space. Rightly did this saying appear over the entrance of the divine Plato's school: "Let no one enter who is ignorant of geometry". [35] (f. 68r. Translation into English in [36]).

Patrizi claimed for the first time in history that geometry is the science of space rather than the *science of continuous magnitudes* assuming that geometry comes first than physics ("mathematica anterior sit quam physiologia") [35] (p. 68r). The idea of priority of mathematics as the science of the space probably pushed Patrizi to publish in 1587 a text on geometry, *Della nuova geometria* [38]. It, though modest from a technical point of view, even mediocre, represents one of the most significant documents for the history of mathematical epistemology [39] (p. 269).

The role attributed to mathematics to define the structure of space, if seen from the modern point of view, is probably the most important novelty introduced by Patrizi; it can be seen as the justification to introduce the structure of space by means of modern mathematics and to classify them: Newton space, Leibniz space, Mach space, Minkowski space time, etc. [3,5]. Patrizi in his time, and also in the early 17th century, was regarded as a leading philosopher, and was translated into English; Giordano Bruno and Henry More read him and were influenced by him, for instance.

Giordano Bruno (1548–1600), a younger contemporary of Patrizi, a member of naturalist Italian school with Bernardino Telesio (1509–1588) and Tommaso Campanella (1568–1639), was a polymath versed in cosmology, astrology, magic, natural philosophy, poetry, and eventually mathematics. This last expertise has been long contested by historians and only recently there has been a reappraisal of his mathematical thought (see for instance [40]). In particular, it is now clear that in Bruno the infinitely large and the infinitely small in natural philosophy are strictly connected with their counterpart in mathematics.

Bruno is known as the philosopher of the infinite, infinite worlds in infinite space, and his conception of space and the cosmos largely influenced his contemporaries; also because of his dramatic condemnation to the stake. He surpassed both Cusanus and Copernicus reaching the fascinating vision of the innumerable worlds that move in an infinite space, that appears as the only container of bodies, as what unites them all. He removed the stoic elements present in Patrizi, that is the existence of a center where to locate the cosmos and any kind of non uniformity, from Patrizi's cosmology. According to him space was essentially an infinite uniform three-dimensional continuum, that ontologically precedes, contains, and receives all things indifferently. His main writing concerning space are the *Cena delle ceneri* [41], *De l'infinito universo e mondi* [42], and *De immenso et innumerabilibus* [43].

Bruno strongly criticized the existence of a center of the universe postulated by Patrizi: "All those who place infinite body and greatness, place neither means nor extreme in that. For whoever says the inane, the void, the infinite aether, does not attribute to it gravity, nor levity, nor motion, nor superior region, nor inferior, nor middle" [42] (vol. 2, p. 36. My translation). Here is somehow recalling Plutarch arguments: "Look at the question broadly. In what sense is the earth middle, and middle of what? [...] for a middle is in a sense itself

a limit, but infinity is a negation of limits [44] (925, p. 24). Additionally, with a different argumentation: "It has already been said several times, and in a better way, that in the infinity the center is everywhere, the term and the principle coincide, the center does not depend on the location of the extreme circumference which in such a space you cannot locate anywhere" [43] (1(1), p. 301); where Cusanus' wording can be recognized: "Hence the fabric of the world will be as if having a center everywhere and its circumference no where" [45] (p. 110).

Similarly Bruno explicitly conceded the uniformity of space which implies and is implied by the uniformity of the universe: "Space must therefore be populated everywhere in the same way, because there would be no reason for something different from the other to be produced in one part of it while the same productive force operates everywhere" [43] (1(2), p. 301).

Bruno's view is that of atomists—an infinite universe and infinite worlds. Indeed it is well documented that Bruno was acquainted with Lucretius' *De rerum natura*, which however most probably was not the main source of his atomism [46] (p. 29). Notably, differently from the atomists, Bruno's space was a plenum, filled with an aether, sometimes named air (but not the common air) or chaos [2], moreover Bruno differently from atomists held that his atoms had only one shape, the spherical one, instead of a great variety.

Bruno often qualified space as a vacuum; actually he used void or vacuum with different acceptations, either as nothing or as a very tiny substance that can be penetrated without opposing any resistance, such as the aether for instance (For the various meanings of vacuum in Bruno see [47] (p. 187)). Differently from common bodies the aether is not a composite of different elements. It is 'spiritual' and devoid of a determined nature, like a unique immense fluid body that maintains its unity:

> In this way we say to be an infinite, that is, an immense aethereal region, in which there are innumerable and infinite bodies, [...], this aether, is not only about these bodies, but still penetrates all of them, and is inherent in everything. We still say vacuum according to meaning, for which we would answer the question, where are the infinite aether and the worlds? [42] (vol. 2, Dialogue 2, p. 32. My translation).

Sometimes Bruno seems to identify space and aether.

Below a quotation which synthesizes Bruno's ideas of space:

> It is therefore not necessary to seek whether in the external sky there are place, vacuum or time; because the general place is one, the immense space that we can freely call void; in which there are innumerable and infinite globes, such as this one in which we live and vegetate. We say such space infinite, because there is no reason, convenience, possibility, sense or nature that must make it finite: in it there are infinite worlds similar to this one, and not different in general from this; because there is neither reason nor defect of natural faculties, I say, as much passive power as active, for which, as there are in this space around us, likewise there are in all the other spaces whose by nature are not different from this one [48] (Dialogue 5, p. 93. My translation).

## 5. The Early Modern Science: Gassendi and Newton

The most important influence of Patrizi and one of the most relevant for the purpose of this entry, however, was on Pierre Gassendi (1592–1655). This is testified in the *Sintagma philosophicum* where Gassendi gave a brief account of earlier ideas about space; when he came to Patrizi, he recognized his debit with him: "about this space, or locus, to which three dimensions length, breadth, and depth belongs, he [Patrizi] propounds nothing other than that what we ourselves have argued about it above" [49] (Vol. 1, *Syntagma philosophicum*, Physica, Section 1, book 3, pp. 246v).

A difference between Patrizi and Gassendi is the latter acceptance of the vacuum and the negation of the existence of a centre. The main difference, however, is in the motivations that push the two scholars to take care of the space. For Patrizi, religious and metaphysical

motivations were dominant, while for Gassendi the stimulus was offered by the study of motion. Both to support his version of Epicurean atomism and to insert himself in the wake of the new Galilean science. Apart from the purely geometrical characteristics of space, such as infinity, isotropy and uniformity, there are two characteristics with a mechanical nature that have fundamental importance for the study of motion: immobility and complete penetrability. With these two characteristics, derived from Patrizi, Gassendi could formulate a more general law of inertia than that of Galileo and substantially coincident with that of Newton [50] (pp. 244–245).

Gassendi reported his conceptions of space and motion in various writings; for instance in the *Syntagma philosophicum*, published posthumously in 1658 in the first two volumes of his *Opera omnia* [49], and in *De motu impressed with a translated motor* (hereinafter more simply *De motu*) of 1642 [51]. A synthesis of Gassendi's thought is found in the *Abrégé de la philosophie de Gassendi* of 1684 [52], and especially in the *Physiologia Epicuro-Gassendo-Charltoniana* of 1654, edited by Walter Charleton [53], the text which mostly contributed to the spreading of the ideas of Gassendi in Europe.

Below Gassendi's conception of space as expressed in the *Syntagma philosophicum*:

The first is that there were immense spaces before God created the World, that these would continue to exist were He perchance to destroy the world; and that of these God has chosen for his own god pleasure the specific region in which to create the World [. . . ].

Secondly, that these spaces are entirely immobile. For it is not the case that If God were to move the World from its present location, that space would follow accordingly and move along with it [. . . ].

Thirdly, that spatial dimensions, without which these spaces would be endlessly open in length, width and depth, as they are immobile, are thus incorporeal, and so have no resistance, or can be penetrated by bodies, or, as it is even commonly said, can coexist with them [49] (Vol. 1, *Syntagma philosophicum*, Physica, Section 1, book 3, pp. 183a,b. Translation into English in [54]).

It is clear from that quotation the similarity between Gassendi and Patrizi.

Isaac Newton (1642–1726), fifty years younger than Gassendi, was indirectly influenced by Patrizi. He knew Gassendi's writings through the treatise by Charleton and was acquainted with Henry More, both were largely influenced by Patrizi. According to [55] (p. 464), he was essentially a Gassendist in the problems of space, time, and existence. Not to forget, however, that Newton was a mathematician and Euclid was inevitably in his brain on a conscious and unconscious level with a natural inclination toward an infinite space. As noticed for Gassendi, a pure mathematical characterization of space was not enough, and Newton had to postulate at least its immovability to speak about an absolute motion and his laws of motion. On this very crucial point Gassendi(Patrizi)'s ideas were fundamental.

If one believed in a continuous progression in the history of ideas, he could see Newton at the top of a pyramid growth in nearly two millennia during which mathematicians, such as Archytas of Tarentum, Democritus, Euclid, Kepler, Galileo, Descartes, Huygens, partially influenced by canonical philosophers of nature and philosophers themselves, enriched their conception of space to make it a suitable receiver of motion. Possibly the progress was not so linear, however Newton was born in an era that allowed him to pick up many of the ideas of ancient mathematicians and philosophers.

Newton exposed his ideas of space in the *De gravitatione* [56,57], a text of controversial dating, with a metaphysical view and, with a more physical mathematical approach, in the Scholium on space and time of the *Principia* (1687); and lastly on the General scholium to the second edition of the *Principia* (1713) [58] (pp. 30–35). Recently a new short piece of writing has been discovered, known as *Tempus and locus*, dated in the 1680s, which presents most succinct statement of how place and time relate to existing things occurs [55,59,60].

Below, a quotation from the Scholium of definitions found in the *Principia*:

Absolute space, of its own nature without reference to anything external, always remains homogeneous and immovable. Relative space is any movable measure or dimension of this absolute space; such a measure or dimension is determined by our senses from the situation of the space with respect to bodies and is popularly used for immovable space [61] (Scholium to definitions; p. 6).

Newton contrasted the relationist view of Descartes with an in depth analysis, in the *De gravitatione*. One of the main criticisms was about the possibility of defining motion because of the difficulty to identify the origin of the trajectories in a universe where everything moves [56] (p. 6).

Certainly Newton devoted much room to the idea of space, he discussed it in depth and justified it on mechanical basis, that is, the idea of absolute space was first of all dictated to justify an absolute motion. After conceiving space and absolute motion, he could think of a principle of inertia to be considered in some way as an empirical principle for which, in absolute space, bodies not subject to forces remain in motion with constant speed and force clearly becomes the cause of the change in motion, not of its maintenance.

The low ontological commitment on the conception of space of the *Principia* and the fact it was introduced in a scholium of the section devoted to definitions, leave a modern reader the impression that Newton did not try to answer the metaphysical question if space is actually absolute or not; on the contrary, he did not even take for granted that such a question was well-posed. His primary aim was instead to define *absolute space*, *absolute time*, and *absolute motion* for applying these concepts and to reveal the roles that they play in solving the problems of mechanics; this was for instance sustained in [62–64] (p. 17).

## 6. Conclusions

Newton, with his absolute space, stands at the culmination of a long series of studies—a dwarf on the shoulders of giants, as he said—started in ancient times. The documents that have come down to us allow a partial reconstruction of the concept of space starting from the fifth century of classical Greece with Democritus and Archita of Taranto. All the themes elaborate. in the following centuries were contained in a nutshell in the writings of Greek philosophers and mathematicians. The atomists prefigured an absolute empty and infinite space. Plato and Aristotle who placed themselves in a relational view imposed their conceptions for a long time. In the Middle Ages, the positions of Aristotle and Plato were reworked and space started to be conceived as a container, even if never empty. A definitive affirmation of space as a container occurred with Patrizi, who saw space as pre-existing to all things, infinite, isotrope, potentially empty though filled with light, the substance that by its nature is closest to space, but especially he saw space as an unmovable and immobile being, which allowed to consider an absolute motion. Moreover Patrizi made the fundamental claim about the priority of geometry in the study of the properties of space (and motion). Gassendi took up Patrizi's positions by inserting them into the atomistic conception he brought into vogue by proposing a vision of space that was substantially that of Newton.

**Funding:** This research received no external funding.

**Institutional Review Board Statement:** Not applicable.

**Informed Consent Statement:** Not applicable.

**Conflicts of Interest:** The author declare no conflict of interest.

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
