# Peer review of "Development of the Concept of Space up to Newton"

_encyclopedia, doi:10.3390/encyclopedia2030104_

Round 1

Reviewer 1 Report

This is a very good entry.

My only objection is to one word. You wrote: Space, is a term that today has many acceptations [9];

For me it is self-contradictory. I understand that "acceptation" is a commonly accepted meaning. If there are many, there is no acceptation at all. 

Author Response

Thanks for your reading and observations,

Reviewer 2 Report

The entry "Development of the Concept of Space up to Newton" takes the framework that was popular with philosophers of physics in the 1980s -- relationism vs. substantivalism -- and seeks to apply it to a number of figures from classical Greece to Newton. There are elements of this piece that are quite good and could be a valuable contribution, but there are a significant number of aspects that require improvement.

The relationist/substantivalist debate was a major part of the discourse around the nature of space in the mid-1710's when Newton defended his account of absolute space and Leibniz attacked it. The author both wants to eliminate contextual aspects of the debate --  "Positions such as the Kantian ones, more oriented toward epistemology, which consider space (with time) one of the forms of sensible knowledge are not discussed. Neither are discussed the points of view of anthropologists or psychologists or theologists (p. 1)." The problem, of course, is that this distinction is contemporary. One cannot draw this distinction among earlier thinkers who saw all of these fields as interconnected. Indeed, there is no way to look at the Newton/Leibniz debate that provides the framework for the author's discussion and see it as in any way removed from theology. Read the Leibniz-Clarke correspondences (Newton provided Clarke's ideas) and the majority of the conversation is about epistemology and theology. Limiting the scope of discussion is necessary, but the author needs to understand that the means of doing so is deeply anachronistic.

Using the substantivalist/relationist debate to structure a global discussion of the nature of space might make sense, but to do so the very first thing you need to do is to formulate a meaningful ontological criterion, that is, establish clearly what does it mean to be a thing? Once you have this condition, you can then look at different accounts of space and see if space according to that account satisfies the criterion. A common ontological criterion is possessing properties. If x has properties, then x must first exist. So, for each view, the question is whether there are properties that belong to space itself and cannot be reduced to relations among things. There are other possibilities that have been proposed as well. If we are going to categorize views of space based on the substantivalist/relationist debate, then the first thing that needs to be done is to make perfectly clear how we categorize according to this distinction and that it is complete and mutually exclusive. To be honest, given where the strengths of this essay really reside, this framing may not be necessary and the author may want to refocus it.

Utilizing the substantivalist/relationist debate as the primary framing mechanism, of course, is anachronistic for the classical views, but could perhaps be enlightening. The central problem for the atomists would be one place which the author could use this categorization to illuminate, but unfortunately doesn't, is the ontological status of the atomists' void. On the one hand, it seems to exist and the atomists seem to posit its reality as a container. It is the real set of places an atom could occupy. But, on the other hand, it seems to have no properties and is described as nothingness. How could nothing be something when nothing is the opposite of something? This is the sort of muddle that could be clarified if the ontological criterion had been worked out.

It would also show how Aristotle's view is difficult to categorize as relationist or substantivalist. The author tries to make the case that it is relationist, yet every spatial point has the property of being sub or superlunary and for sublunary points has the property of being within the natural resting place of some sort of matter. Is that sufficient to endow each with a property and therefore to make Aristotelian space substantival? But there is no way to tell one, say, water-resting point from another. Is that sufficient to make it relational? Perhaps the problem is that the 1710's framing just doesn't fit well here.

The other concern with the classical section is the Eurocentrism. The Greeks were not the only classical civilizations with interesting views of space. John North's wonderful book Cosmos is a valuable resource in looking at East and South Asian, Pre-Columbian American, and other classical civilizations who had non-trivially complex views of space. These would influence each other in the golden age of Islam wherein spherical trigonometry would be applied to the heavens. Surely, this is mathematical, yet it had a religious element -- again, these are not really separable in the historical context.

The move to the science of the Scholastic period was deeply influenced by Plato and Aristotle -- why? Because of the influence of the neo-Platonist and Thomist debate within the Church. Again, this shows how the story cannot be neatly separated into physics vs. non-physics.

One of the stronger sections is that which examines the Middle Ages and Renaissance figures, although it would be good to do more scholarship and work through them in greater scholarly detail. Work through, with more references to primary sources, what exactly the view of space was for each and how it built upon, but deviated from those who went before. More needs to be done on Copernicus and Crescas. The Patrizi section is quite good, but a bit more on Bruno would be helpful. Gassendi section is also very nice. To be honest, this essay could be refocused to examine this period and it would make it a much better essay. It would explain why we only need to look at the Greeks among the classical civilizations – because they are the prelude to this period – and it could still end with Newton because that is where the process does historically go and it means that the author does not need to bolster the Newton section – which is thin given the incredible amount of scholarship there is on Newton’s view of space. This is the strongest part of the essay and ought to be turned into the point of the essay. It is also a time period that is woefully and wrongly overlooked by many scholars interested in the history of space. That would make this piece much more valuable to the discourse community.

There was one bit of English in this section that MUST be addressed. “the Jew philosopher Hasday Crescas” should read “the Jewish philosopher Hasday Crescas.” To use “Jew” as an adjective or verb in many English-speaking communities is antisemitic.

The one thing that is lost in this section is the framing of the relationist/substantivalist debate. But, to be honest, it is not missed. My suggestion would be to lose that framing altogether. Make the Greek section a prelude which provides enough detail (as it almost does now) to make sense of the Middle Ages and Renaissance debates and focus on those figures in more detail. That is the real strength of the piece as it is written and would contribute most to the discussion. Strengthen the strength and jettison that which is unnecessary. It would require little in terms of revision, more of a refocusing which would make this a tighter, more interesting piece.

Which brings us to Newton. From the title of the piece, the assumption is that this will be a closed interval including Newton, not an open interval in which the limit of the discussion approaches Newton. Of course, it was a good decision on the part of the author not to go into Newton as that is at least an entry unto itself. The literature is robust and certainly more than this piece could handle given its focus really lays elsewhere.

There is a lot to like in this piece and I think that with some rethinking it could be a very valuable contribution to the history of space.

Author Response

Thanks for your very detailed and acute criticisms. Some of them would require to restructure completely the paper, which is not practicable for me at the moment and which I probably do not want to do. I must confess however that some of your comments are not very clear to me. Hope my corrections will meet them. In blue the answer to your observations; in the PDF major corrections are evidenced in red.

The entry "Development of the Concept of Space up to Newton" takes the framework that was popular with philosophers of physics in the 1980s -- relationism vs. substantivalism -- and seeks to apply it to a number of figures from classical Greece to Newton. There are elements of this piece that are quite good and could be a valuable contribution, but there are a significant number of aspects that require improvement.

The relationist/substantivalist debate was a major part of the discourse around the nature of space in the mid-1710's when Newton defended his account of absolute space and Leibniz attacked it. The author both wants to eliminate contextual aspects of the debate --  "Positions such as the Kantian ones, more oriented toward epistemology, which consider space (with time) one of the forms of sensible knowledge are not discussed. Neither are discussed the points of view of anthropologists or psychologists or theologists (p. 1)." The problem, of course, is that this distinction is contemporary. One cannot draw this distinction among earlier thinkers who saw all of these fields as interconnected. Indeed, there is no way to look at the Newton/Leibniz debate that provides the framework for the author's discussion and see it as in any way removed from theology. Read the Leibniz-Clarke correspondences (Newton provided Clarke's ideas) and the majority of the conversation is about epistemology and theology. Limiting the scope of discussion is necessary, but the author needs to understand that the means of doing so is deeply anachronistic.

You are completely right of course, thanks for your criticism anyway. It is virtually impossible to avoid any form of anachronism but one can attempt to reduce it. I changed a little the specification of the object of the paper that gives better the meaning I really wanted to say. Hope this will reduce a little the suspect of anachronism.

Using the substantivalist/relationist debate to structure a global discussion of the nature of space might make sense, but to do so the very first thing you need to do is to formulate a meaningful ontological criterion, that is, establish clearly what does it mean to be a thing? Once you have this condition, you can then look at different accounts of space and see if space according to that account satisfies the criterion. A common ontological criterion is possessing properties. If x has properties, then x must first exist. So, for each view, the question is whether there are properties that belong to space itself and cannot be reduced to relations among things. There are other possibilities that have been proposed as well. If we are going to categorize views of space based on the substantivalist/relationist debate, then the first thing that needs to be done is to make perfectly clear how we categorize according to this distinction and that it is complete and mutually exclusive. To be honest, given where the strengths of this essay really reside, this framing may not be necessary and the author may want to refocus it.

Utilizing the substantivalist/relationist debate as the primary framing mechanism, of course, is anachronistic for the classical views, but could perhaps be enlightening. The central problem for the atomists would be one place which the author could use this categorization to illuminate, but unfortunately doesn't, is the ontological status of the atomists' void. On the one hand, it seems to exist and the atomists seem to posit its reality as a container. It is the real set of places an atom could occupy. But, on the other hand, it seems to have no properties and is described as nothingness. How could nothing be something when nothing is the opposite of something? This is the sort of muddle that could be clarified if the ontological criterion had been worked out.

What is the nothing of atomist has been a long debate, that possibly will not end because their writings reduce to some fragments. Here it is assumed the standard interpretation for which nothing is really something, which is classical. I tried to add something. Any way I do not dare to give my own interpretation.

It would also show how Aristotle's view is difficult to categorize as relationist or substantivalist. The author tries to make the case that it is relationist, yet every spatial point has the property of being sub or superlunary and for sublunary points has the property of being within the natural resting place of some sort of matter. Is that sufficient to endow each with a property and therefore to make Aristotelian space substantival? But there is no way to tell one, say, water-resting point from another. Is that sufficient to make it relational? Perhaps the problem is that the 1710's framing just doesn't fit well here.

In the paper I assumed Aristotle had a relationalist position, even though from certain point his space has something of absolute, because at the least the earth is at rest, and also space is at rest, as is the case for absolute space.

The other concern with the classical section is the Eurocentrism. The Greeks were not the only classical civilizations with interesting views of space. John North's wonderful book Cosmos is a valuable resource in looking at East and South Asian, Pre-Columbian American, and other classical civilizations who had non-trivially complex views of space. These would influence each other in the golden age of Islam wherein spherical trigonometry would be applied to the heavens. Surely, this is mathematical, yet it had a religious element -- again, these are not really separable in the historical context.

I browsed the book by John North. I found here very interesting hints to Mediterranean civilization, and considered it worthy to be cited. However my history will start from ancient Greek and not European civilization are eluded, for many reasons, of space, of time and of competence. The text is however a little changed and North’s book is cited.

The move to the science of the Scholastic period was deeply influenced by Plato and Aristotle -- why? Because of the influence of the neo-Platonist and Thomist debate within the Church. Again, this shows how the story cannot be neatly separated into physics vs. non-physics.

One of the stronger sections is that which examines the Middle Ages and Renaissance figures, although it would be good to do more scholarship and work through them in greater scholarly detail. Work through, with more references to primary sources, what exactly the view of space was for each and how it built upon, but deviated from those who went before. More needs to be done on Copernicus and Crescas. The Patrizi section is quite good, but a bit more on Bruno would be helpful. Gassendi section is also very nice. To be honest, this essay could be refocused to examine this period and it would make it a much better essay. It would explain why we only need to look at the Greeks among the classical civilizations – because they are the prelude to this period – and it could still end with Newton because that is where the process does historically go and it means that the author does not need to bolster the Newton section – which is thin given the incredible amount of scholarship there is on Newton’s view of space. This is the strongest part of the essay and ought to be turned into the point of the essay. It is also a time period that is woefully and wrongly overlooked by many scholars interested in the history of space. That would make this piece much more valuable to the discourse community.

There was one bit of English in this section that MUST be addressed. “the Jew philosopher Hasday Crescas” should read “the Jewish philosopher Hasday Crescas.” To use “Jew” as an adjective or verb in many English-speaking communities is antisemitic.

I added something about Cresca, and Bruno. For the last I was a little embarrassed; apart from reason of space I think that Bruno is much more interesting for his cosmological ideas than for those  regarding strictly space and because he was read by most scholar in the 17th century. Anyway, I added something about Bruno as well. Thanks for the correction of English.

The one thing that is lost in this section is the framing of the relationist/substantivalist debate. But, to be honest, it is not missed. My suggestion would be to lose that framing altogether. Make the Greek section a prelude which provides enough detail (as it almost does now) to make sense of the Middle Ages and Renaissance debates and focus on those figures in more detail. That is the real strength of the piece as it is written and would contribute most to the discussion. Strengthen the strength and jettison that which is unnecessary. It would require little in terms of revision, more of a refocusing which would make this a tighter, more interesting piece.

Which brings us to Newton. From the title of the piece, the assumption is that this will be a closed interval including Newton, not an open interval in which the limit of the discussion approaches Newton. Of course, it was a good decision on the part of the author not to go into Newton as that is at least an entry unto itself. The literature is robust and certainly more than this piece could handle given its focus really lays elsewhere.

There is a lot to like in this piece and I think that with some rethinking it could be a very valuable contribution to the history of space.

Reviewer 3 Report

The manuscript "Development of the concept of space up to Newton
by Danilo Capecchi is an interesting overview of te development of concept of space from presocratic natural philosopher, atomists Democritus and Epicurus, Stoics, Plato, Aristotle, Middle Ages, Renaissance,Francesco Patrizi, Giordano Bruno up to Gassendi and Newton. The article is well and clearly writen and will be interesting for the readers. I recommend its publication in present form after the correction of a couple of misprints.  

p. 8 l. 6 from the end: quire clear --> quite clear
p. 9. l 2 from the end. atimists --> atomists
p. 12, l. 4: off --> of
p. 12, l. 6: immagine  --> imagine

Author Response

Thanks for your reading and comments

Reviewer 4 Report

The article gives a fine survey that is on the whole accurate. It could be improved by more up-to-date and useful literature references.

For instance, should Euclidean geometry be taken to be based on a concept of space, or is it solely about figures, with no ambient space presupposed in any essential way? Vincenzo De Risi has argued the latter in many recent works; see for example, Vincenzo De Risi (ed.), Mathematizing Space: The Objects of Geometry from Antiquity to the Early Modern Age, 2015. The author refers to this issue only very briefly and vaguely and through an outdated reference to Edward Grant.

The relative space alternative to Newtonian absolute space is not given adequate coverage. As the author states in one sentence:

"subsequent development of physics, has shown that the resistance of Leibniz (and Huygens), supporter of a relationalist views, intuitively well founded but supported by inadequate arguments, was actually justified, see for instance [10,11]."

This is essentially correct, except I would not say that the arguments for relative (or "relationalist") notions of space were any more "inadequate" than any of the other arguments discussed in the article. In particular, Newton's arguments for absolute space were obviously inadequate too, let alone the arguments of earlier thinkers.

The relative space view deserves a fuller hearing. Indeed, not only Leibniz and Huygens but also Descartes were relativists about space. It would be appropriate to cover this in the article, and to reference some appropriate recent scholarship instead of the rather strange and obscure references [10,11] the author provides.

The abstract starts: "Space is one of the concepts that inevitably presents itself to all men from birth. Its perception, even for the common man, has changed profoundly in the course of history." The article is not really about this, but more about traditional debates in philosophical literature. If the author really wants to take the broader cognitive view alluded to in the abstract, an interesting reference might be Matthias Schemmel, Historical Epistemology of Space: From Primate Cognition to Spacetime Physics, 2015.

Author Response

Thanks for your valuable comments and criticisms. In blue the answer to your observations; in the PDF major corrections are evidenced in red.

The article gives a fine survey that is on the whole accurate. It could be improved by more up-to-date and useful literature references.

For instance, should Euclidean geometry be taken to be based on a concept of space, or is it solely about figures, with no ambient space presupposed in any essential way? Vincenzo De Risi has argued the latter in many recent works; see for example, Vincenzo De Risi (ed.), Mathematizing Space: The Objects of Geometry from Antiquity to the Early Modern Age, 2015. The author refers to this issue only very briefly and vaguely and through an outdated reference to Edward Grant.

Thanks for the suggestion of De Risi’s book, that I did not know; I read and appreciate it a lot. What De Risi sustains, i.e. that at the beginning Greek geometry was a geometry of figure and not of space, this is true, at least I agree with him. This however does not imply that Greek mathematicians were fully aware of their treatment and did not thought about an external space.

The relative space alternative to Newtonian absolute space is not given adequate coverage. As the author states in one sentence:

"subsequent development of physics, has shown that the resistance of Leibniz (and Huygens), supporter of a relationalist views, intuitively well founded but supported by inadequate arguments, was actually justified, see for instance [10,11]."

This is essentially correct, except I would not say that the arguments for relative (or "relationalist") notions of space were any more "inadequate" than any of the other arguments discussed in the article. In particular, Newton's arguments for absolute space were obviously inadequate too, let alone the arguments of earlier thinkers.

The relative space view deserves a fuller hearing. Indeed, not only Leibniz and Huygens but also Descartes were relativists about space. It would be appropriate to cover this in the article, and to reference some appropriate recent scholarship instead of the rather strange and obscure references [10,11] the author provides.

 I know that I gave only a hint to the relative notion of space. This is a choice because my purpose was to show how the idea of absolute space emerged in the history. For what the citation [10, 11] is concerned you are perfectly right, simply they are located in the wrong position. I added two more citations concerning the modern position about absolute and relative space. I chose not to give much space to Descartes, though he was a supporter of a relativist view of space, for the sake of space. I however added something about him in the last section.

The abstract starts: "Space is one of the concepts that inevitably presents itself to all men from birth. Its perception, even for the common man, has changed profoundly in the course of history." The article is not really about this, but more about traditional debates in philosophical literature. If the author really wants to take the broader cognitive view alluded to in the abstract, an interesting reference might be Matthias Schemmel, Historical Epist

You are perfectly right; I changed a little the starting sentence of the abstract that betrays my previous project. I browsed Schemmel book, I knew the title but not the content. Thanks for reporting it to me.

Round 2

Reviewer 2 Report

"Development of the Concept of Space up to Newton" offers a discussion of the nature of space from classical Greece up through the post-Renaissance era. While there is a significant amount of scholarly attention paid in the literature to the Greeks and less, albeit plenty to the Roman atomists, the period between these thinkers and Galileo tends to be neglected. This paper fills that gap.

The scholarship is appropriate for Encyclopedia, in depth enough to give a good sense of the various arguments but not so much that one gets lost in the details. Rather, there is a clear intention on the pat of the author to maintain a holistic sense of where each thinker fits into the larger historical trajectory. We see each tree in focus without ever losing the sense of the forest.

This revision is an improvement on the initial submission. The author took suggestions seriously and made the requested additions, extensions, and changes. I believe this version will be publishable after a good and thorough copy editing. There are still some issues with the English, but the content and structure are successful.

Author Response

Thanks again for your suggestions

Reviewer 4 Report

Revisions seem fine. 

Author Response

thanks again for your suggestions